# Ultrasonic Bending Vibration-Assisted Purification Experimental Study of 7085 Aluminum Alloy Melt

**DOI:** 10.3390/ma15103598

**Published:** 2022-05-18

**Authors:** Chen Shi, Jiangnan He, Hua Liao, Daheng Mao

**Affiliations:** 1Light Alloy Research Institute, Central South University, Changsha 410083, China; 203812049@csu.edu.cn (J.H.); liaohua@csu.edu.cn (H.L.); mdh@csu.edu.cn (D.M.); 2State Key Laboratory of High Performance Complex Manufacturing, Central South University, Changsha 410083, China

**Keywords:** 7085, aluminum melt, ultrasonic bending vibration, inclusions, purification

## Abstract

Aiming at the problem that melt inclusions in the casting process of 7085 aluminum alloy seriously affect the ingot quality, this study introduces ultrasonic bending vibration into the melt of the launder in the semi-continuous casting process of 7085 aluminum alloy and investigates the online purification effect of ultrasonic bending vibration on the melt of 7085 aluminum alloy through a metallographic analysis, SEM analysis, and EDS energy spectrum analysis. The results show that, under the action of the ultrasonic, the inclusions in the aluminum melt are transformed from a large number of elongated large inclusions with a size of more than 50 μm, and granular inclusions with a size of about 5–15 μm, into a small amount of smaller than 30 μm point-like small inclusions. In addition, the average area ratio of inclusions in the melted sample was reduced from 3.835 (±0.05)% to 0.458 (±0.05)%, and the residual refining agent in the aluminum melt was effectively removed. It was also found that under the action of ultrasonic bending vibration, the tiny inclusions in the melt aggregate with each other, and interact with the residual refining agent in the melt to further grow, and are attached to the inner surface of the ceramic cavity channel to be removed.

## 1. Introduction

7085 aluminum alloy is an ultra-high-strength heat-treatable reinforced aluminum alloy with a high-specific strength and specific stiffness, and an excellent processing performance. It has been widely used in aerospace, rail vehicles, and other fields [1,2,3,4,5]. In the casting process, the inclusions and gases in the melt have a great influence on the quality of the 7085 aluminum alloy ingot. Therefore, to prepare a high-quality 7085 aluminum alloy ingot, purifying the melt is of urgent importance.

At present, domestic and foreign methods for metal melt purification mainly include the purging method, filter purification method, and refiner purification method [6], and have been applied in production, but there are still certain drawbacks. The purging method is mainly used for gas removal, and the efficiency of removing non-metallic inclusions is low [7,8]. A foam ceramic filter purification method can effectively remove non-metallic inclusions, but a skeleton material with a certain impact resistance and suitable branch density must be selected [9,10,11]. A refiner purification method can remove inclusions, but too much refiner will cause the pollution to the melt [6,12,13].

Ultrasonic-assisted melt purification is a technology that applies ultrasonic vibration to the metal melt to assist in purifying the melt, which not only solves the problems of a single purification target and low purification efficiency, but most importantly is green and does not cause large secondary pollution to the environment and the melt [14,15,16], which has a broad application prospect.

Liu Jingshan et al. [17] studied the effect of ultrasonic vibration on the organization and properties of 5356 aluminum alloy and prepared 5356 aluminum alloy by changing the insertion depth of the variable amplitude rod and the ultrasonic treatment time, among other things. The aluminum alloy was uniformly organized after ultrasonic treatment, and most of the oxides were removed. Dong Chengshu et al. [18] studied the effect of ultrasonic stirring on the degassing and debridement of aluminum alloy melt under different parameters and found that with the increase of ultrasonic time and power, the pores and inclusions in the aluminum alloy were greatly reduced, while the mechanical properties of the material were greatly improved. Li Haoyu et al. [19] studied the agglomeration effect of ultrasound on inclusions, and under the action of the ultrasonic standing wave field, the inclusions moved to the acoustic pressure node and acoustic pressure ventral node perpendicular to the propagation direction. The inclusions are formed under the action of the acoustic pressure interaction force as well as the side force, and the inclusions will float or sink in the subsequent resting time to achieve the purpose of removing the inclusions.

In summary, ultrasonic-assisted purification as a new melt purification technology is being paid more and more attention and researched by researchers, but ultrasound is still inadequate in purifying large volume melts. Xu et al. [20] studied the effect of ultrasound on the purification and degassing of different volume melts, and the results showed that the ultrasonic degassing rate of large volume melts was significantly lower than that of small volume melts. The online ultrasonic purification of flow tank melt can solve the problem of poor purification effect of large volume melt, but the traditional straight rod type ultrasonic waveguide rod mainly exports the longitudinal vibration along the axial direction, and the vibration exported along the radial direction (flow direction of launder melt) is very little, which is not conducive to the effective purification of launder melt. In this study, an L-type high-temperature resistant ultrasonic waveguide rod [21,22,23,24] was developed to introduce ultrasonic bending vibration into the melt of 7085 aluminum alloy in the semi-continuous casting process to enhance the high-frequency vibration output along the flow direction of the melt in the launder, and an experimental study of ultrasonic bending vibration-assisted online purification of the melt of 7085 aluminum alloy was conducted.

## 2. Materials and Methods

### 2.1. Experimental Materials and Equipment

The composition of 7085 aluminum alloy used in the experiments is shown in Table 1. The ultrasonic transducer generates high-frequency longitudinal vibration, which is transformed into high frequency bending vibration by the L-shaped waveguide rod and then introduced into the molten 7085 aluminum alloy in the launder. In addition, a ceramic cavity with a honeycomb channel made of special ceramic material (the diameter of the channel is 6 mm) is arranged in the launder before and after the ultrasonic rod, and the molten 7085 aluminum alloy in the flow tank passes through the front ceramic cavity, the ultrasonic bending vibration rod, and the rear ceramic cavity, and then flows into the crystallizer. Figure 1 is a two-dimensional diagram and a physical photos of the ultrasonic purification device for online purification of the launder.

### 2.2. Experimental Process

This purification experiment was carried out during the semi-continuous casting and forming process of 7085 ingots with a size of Φ830 mm × 5000 mm, and the online purification treatment of the aluminum melt in the launder before entering the crystallizer was carried out. First, the 7085 aluminum melt is subjected to in-furnace refining (refining agent), out-of-furnace compound refining (argon + CCl_4_), and two-stage ceramic plate filtration (first-stage 30 mesh, second-stage 60 mesh); then, the filtered the 7085 aluminum melt flows into the launder, flows through the front ceramic cavity, the ultrasonic bending vibration rod, and the rear ceramic cavity of the ultrasonic-assisted purification device in turn, and then flows into the crystallizer. After the semi-continuous casting begins, the ultrasonic is turned on. The frequency is 23.8 KHz, the ultrasonic amplitude is 6 μm, and the depth of the ultrasonic rod inserted into the melt is 65 mm. The test site is shown in Figure 2.

During the experiment, 4 groups of melt samples were randomly taken from the front and rear aluminum melts of the ultrasonic purification device (three groups of samples were taken when the ultrasonic action was turned on, and the other group of samples was taken when the ultrasonic action was not turned on). The sampling method is as follows: first remove the melt surface film of the launder, then use a sampling spoon to sample from the inside of the melt and perform subsequent test analysis after the melt sample is air-cooled and solidified. At the same time, after the experiment, the residual aluminum liquid in the honeycomb channels of the front and rear ceramic cavities was tested and analyzed.

### 2.3. Test and Analysis Methods

(1)Microstructure analysis of launder melt samples before and after ultrasonic purification

Use a wire cutting machine to cut the front and rear four samples taken in half along the axis. The samples were ground and polished in the MP-2B polishing machine (Milland, Changzhou, China). The samples were polished with 600#, 800#, 1000#, 1500#, and 2000# sandpaper in turn, then polished with polishing paste, and finally cleaned with absolute ethanol and then dried. The area fraction values of inclusions in the metallographic pictures were quantified using a metallographic microscope (FCM-2000 W, Leeb, Chongqing, China) with 10 randomly selected fields of view for each sample combined with ImageJ (V1.8.0) image analysis software, and the morphology, size, and distribution of inclusions in the samples were observed using a fully automatic scanning electron microscope (TESCAN MIRA C7301, TESCAN, Shanghai, China), followed by scanning the inclusions composition and content using EDS.

(2)Microstructure analysis of the residual metal in the ceramic cavity and the inner surface of the cavity channel

The solidified residual melt sample (bar shape) was taken out of the ceramic cavity channel, and its cross-section was ground and polished in the MP-2B polishing machine. The morphology, size, and distribution of inclusions in the residual aluminum alloy sample were observed using fully automatic scanning electron microscopy (TESCAN MIRA C7301, TESCAN, Shanghai, China), and the composition and content of inclusions were observed using EDS scanning, while the elemental composition analysis was performed on the surface of the residual aluminum alloy sample and the inner surface of the channel of the ceramic cavity.

(3)Mechanical properties testing of launder melt samples

The tensile mechanical properties of the melt samples before and after purification were tested in a MTS 810 universal material testing machine (MTS, MN, USA) at a strain rate of 2 mm/min. Three tensile specimens were taken out of the melt-solidified samples before and after purification, and the dimensions of the tensile specimens were shown in Figure 3. The ultrasonic-assisted purification was investigated by comparing the average values of the three parallel samples before and after purification. The effect on the melt solidification microstructure and properties, and then the macroscopic effect of ultrasonic-assisted purification of melt inclusions was obtained.

## 3. Results and Analysis

### 3.1. Metallographic Analysis of Inclusions in Melt Samples before and after Purification Device When Ultrasonic Is not on

As shown in Figure 4, the microscopic metallographic analysis (100 times magnification) was carried out on the melt samples before and after the purification device when the ultrasonic wave was not turned on. It can be seen from the figure that when the ultrasonic wave is not turned on, after the aluminum melt flows through the purification device, the inclusion content does not change significantly, and the shape and size of the inclusions are not significantly different, which shows that although the ceramic cavity in the purification device is placed. However, the diameter of the honeycomb channel in the ceramic cavity is large (6 mm), which cannot filter and purify the aluminum melt. Therefore, without turning on the ultrasound, the aluminum melt flowing through the ceramic cavity is not effectively purified.

### 3.2. Metallographic Analysis of Inclusions in Melt Samples before and after Ultrasonic Purification

A microscopic metallographic analysis (100 times magnification) of the melt samples before and after ultrasonic purification was carried out using an optical metallographic microscope. Ten randomly selected fields of view from each specimen were used to perform statistical calculations, and the unit area of inclusions in metallographic pictures was quantified and calculated by ImageJ image analysis software, and the ratio of the area of inclusions to the area of the captured metallographic pictures was obtained.

As Figure 5 illustrates, there are many large sizes of long inclusion defects in the melt specimens prior to ultrasonic purification, inclusions greater than 50 μm, and inclusions with a length of 239.60 (±2) μm, as shown in Figure 5a at point 1, and 283.03 (±2) μm shown in Figure 5c at point 2. The quantitative analysis of metallographic images by ImageJ image analysis software shows that the average area of inclusions in the specimen is 3.835 (±0.05)%, while no large-sized inclusions are found in the melt specimen after ultrasonic purification, and the inclusions are less than 30 μm in size. Additionally, the number of inclusions is significantly reduced, and the average area of inclusions is reduced to 0.458 (±0.05)%, which shows that ultrasonic bending vibration has a significant effect on the online purification of 7085 melt.

### 3.3. Scanning Electron Microscopy Analysis of Inclusions in Melt Samples before and after Ultrasonic Purification

Fully automatic scanning electron microscopy was used to observe the molten specimens in the areas where no inclusions were found in the metallographic pictures, and further microstructure analysis was performed by randomly selecting the field of view and magnifying it 500 times to obtain SEM micrographs. The SEM electron microscopy images of the specimens are shown below.

From the SEM photographs of the melt specimens in Figure 6, it can be seen that more granular aggregates of small inclusions can be clearly observed in the melt specimens before ultrasonic purification, while it is difficult to see such small inclusions in the melt specimens after ultrasonic purification.

The above tissues were further magnified for observation and analysis using a scanning electron microscope, and Figure 7 shows the 3000 times magnification of the local specimen tissues before and after purification.

As Figure 7 illustrates, in the specimens before ultrasonic treatment, the inclusions defects mainly consist of some inclusions and gas pores with the size of about 5–15 μm, and there are multiple small size inclusions gathered together. In addition, Figure 7a–c illustrates that the surface burrs of the inclusions defects are more frequent and angular. The inclusions defects in the specimens after ultrasonic treatment are also composed of inclusions and gas pores, which are small in size at about 1–5 μm. It is generally believed that inclusions and gas are interdependent, that there must be gas in the melt, and that more inclusions equal more gas pores [25]. This indicates that after online ultrasonic treatment, the inclusions and hydrogen content in the melt specimens are significantly reduced.

The results of our EDS energy spectrum analysis of the region of points 1–10 in Figure 7 are shown in Table 2. An analysis of the elements and content of the points in the table shows that the inclusions at each point appear as O and other non-alloy component elements, which can indicate that the inclusions in the melt samples are mainly composed of oxides, while inclusions such as Cl, Na, Ca, Rb, and other refiner component elements, were found in the inclusions of the melt specimens before ultrasonic treatment (points 1–7). However, they were not found in the inclusions (points 8–10) of the melt specimens after ultrasonic treatment.

### 3.4. Analysis of Cross-Sectional Microstructure and Inclusions Composition of Residual Aluminum Melt Specimens Inside the Ceramic Cavity

After the completion of the semi-continuous casting test, the aluminum melt specimens (rod-shaped) remaining in the middle of the front and rear ceramic cavities were taken, and the cross-section was ground and polished, and then the microstructure of the cross-section was observed under a fully automatic electron microscope. The microstructure is as shown below.

The SEM images show that no obvious inclusions and pores are found in the center of the residual aluminum melt specimens inside the ceramic cavity, and only a few pores are seen in the transition region, while pores and inclusions exist in the edge region. A comparison of Figure 8 and Figure 9 of the edge area can be found, before the ceramic cavity inside the residual aluminum melt sample edge area exists more pores and inclusions than after the ceramic cavity, and they are obvious from the edge to the heart of the long strip of inclusions. A comparison of the microscopic morphology of the central, transition, and edge zones of the residual aluminum melt specimens inside the ceramic cavity shows that the melt inclusions tend to move toward the wall surface inside the ceramic cavity after the ultrasonic treatment.

The impurity material defects in the front and rear ceramic cavity residual aluminum alloy specimens were further magnified under a fully automated electron scanning microscope and analyzed by EDS energy spectroscopy, and the results are shown in Figure 10.

As can be seen from Figure 10, the inclusions of the residual melt in the ceramic cavity are accompanied by the porosity, and from the composition analysis of EDS, it can be seen that the inclusions of the residual melt in the ceramic cavity contain significantly more composition elements than the launder melt samples (as shown in Table 2). In the elemental composition of inclusions, in addition to the O element, there are also refining agents such as K, Ca, and Cl. This is mainly due to the effect of ultrasonic cavitation nuclei in the ceramic cavity to agglomerate the tiny inclusions in the melt and interact with the residual refining agent in the melt, resulting in the size of the inclusions becoming larger, and in the ceramic cavity channel it floats or sinks to the inner surface of the ceramic cavity and is attached to the inner surface of the ceramic cavity so that it does not flow to the crystallizer with the melt.

### 3.5. Ceramic Cavity Internal Residual Aluminum Melt Specimen Surface and Ceramic Cavity Channel Internal Surface Composition Analysis

The EDS energy spectra of the surface of the residual aluminum sample in the ceramic cavity, and the inner surface of the cavity channel, were analyzed as shown in Table 3, and the composition points were taken as shown in Figure 11. It can be seen from Table 3 that, like the inclusions in the cross-sectional edges of the residual melt sample in the ceramic cavity, Cl, K, Na, and Ca were also found on the surface of the residual aluminum melt sample in the ceramic cavity and on the inner surface of the ceramic cavity channel, indicating that the tiny inclusions in the melt aggregated under the action of ultrasound and interacted with the residual refining agent in the melt to adhere to the inner surface of the ceramic cavity channel in close proximity.

### 3.6. Effect of Ultrasonic Purification Treatment on Mechanical Properties of Solidified Samples of Aluminum Melt in Launder

Table 4 shows the tensile mechanical properties of the solidified samples of aluminum melt in the launder before and after ultrasonic purification. After comparison, it was found that after ultrasonic purification, the tensile strength of the solidified sample of aluminum melt in the launder increased by 9.5%, the yield strength increased by 4.2%, and the elongation increased by 57.7%, which indicated that the solidified samples of aluminum melt before ultrasonic purification contained more inclusions, which resulted in premature fracture of tensile samples, resulting in low tensile mechanical properties. Moreover, the aluminum melt after ultrasonic purification was effectively purified, the inclusions were significantly reduced, and the tensile mechanical properties of the aluminum melt solidification sample were high. This further shows that the ultrasonic bending vibration is effective for the online purification of 7085 melt.

## 4. Discussion

After the ultrasonic bending vibration is introduced into the 7085 aluminum melt, a nonlinear positive and negative alternating sound pressure is formed in the aluminum melt resulting in a cavitation effect, which in turn acts on the gas and inclusions in the melt to be discharged. In the ultrasonic-assisted purification device used in this study, a ceramic cavity made of special ceramic material with honeycomb channels is arranged before and after the ultrasonic rod in the launder, and the cavitation effect of ultrasonic bending vibration is used to effectively purify the aluminum melt flowing through the ceramic cavity. When the ultrasonic vibration propagates in the aluminum melt, the energy dispersion caused by the diffusion of the sound beam, and the scattering effect of the medium and the absorption effect of the melt itself [26,27], the energy attenuation will appear with the increase of the propagation distance. When the ultrasonic sound pressure is attenuated below the cavitation threshold for nucleated cavitation, the cavitation effect will not occur, which means that inclusions in the melt in this region cannot be effectively purified. Relevant studies have shown that the threshold for nucleated cavitation in aluminum melt is 0.1 MPa [28], and the relationship between ultrasonic sound intensity and sound pressure is as follows:(1)$$\;I=\frac{P^{2}}{\rho c}$$

In the formula, *I* represents the sound intensity W/m^2^, *P* represents the sound pressure Pa, *ρ* represents the medium density kg/m^3^, and *c* represents the sound speed m/s.

According to the above formula, it can be deduced that the ultrasonic sound intensity must be greater than or equal to 0.058 W/cm^2^ if nuclear cavitation is to be generated in the aluminum melt, and the attenuation law of ultrasonic sound intensity can be expressed by the following formula [29]:(2)$$I=I_{0}e^{-2\alpha x}$$

In the formula, *I_0_* represents the sound intensity at the initial position, *α* represents the attenuation coefficient, and *x* represents the distance from the initial position.

Zhang Wu et al. [30] calculated that the attenuation coefficient of power ultrasonic propagation in aluminum melt is 0.01103 mm^−1^, and the initial sound intensity of ultrasonic vibration used in this study at the starting position is 0.80 W/cm^2^. Substituting into the above formula shows that when the distance from the ultrasonic rod is 119 mm, and the ultrasonic sound intensity is 0.058 W/cm^2^, so the position of the ceramic cavity in this study is within 119 mm from the ultrasonic rod, which is in the effective “cavitation zone”, as shown in Figure 12a. The reasons for the purification of the 7085 aluminum melt flowing through the effective “cavitation zone” include the following three aspects:(1)When the aluminum melt flows into the ceramic cavity, the aluminum melt enters the effective “cavitation zone” of ultrasonic vibration. The inclusions in the aluminum melt are usually not wetted with the metal melt, and there are many inclusions on the surface of the inclusions. In the concave area, there are many small air bubbles in these concave areas. These small air bubbles can easily become “cavitation nuclei”, which expand and become larger under the negative pressure of ultrasonic vibration, and float up to the inner surface of the ceramic cavity with inclusions to be discharged. In addition, the inclusions will collide and grow up under the effect of ultrasonic acoustic flow, and they will float or sink nearby and adhere to the inner surface of the ceramic cavity to be removed [31,32]. As shown in Figure 12b.(2)On the other hand, as shown in Figure 12c, there are a large number of microscopic uneven areas on the inner surface of the honeycomb channel of the ceramic cavity. The aluminum melt does not wet with the inner surface of the channel, and the microscopic unevenness on the inner surface of the channel. There are tiny bubbles attached to the uneven area. Under the action of ultrasonic cavitation, the tiny bubbles expand and become larger, and the hydrogen content in the bubble is lower than that in the melt [31], which promotes the hydrogen and tiny “cavitation nuclei” around the bubble into the bubble and are fixed on the inner surface of the ceramic cavity. The effective purification of inclusions is achieved [25], as shown in Figure 12d.(3)In this study, the 7085 aluminum melt has been refined in the furnace (refining agent), compound refining outside the furnace (argon + CCl_4_, and double-stage ceramic plate filtration (30 mesh in the first stage and 60 mesh in the second stage) before entering the launder. The large-scale inclusions in the melt have been effectively removed, but through the aforementioned experimental analysis, it is found that there are still inclusions and residual refining agents in the aluminum melt that has just flowed into the launder. After the aluminum melt flows through the ultrasonic-assisted purification device, the fine inclusions in the aluminum melt aggregate and grow, and then can interact with the residual refining agent so that the residual refining agent in the melt can be effectively removed, thereby avoiding the residual refining agent entering the crystallizer melt and causing solidification defects.

## 5. Conclusions

7085 aluminum alloy is widely used in aerospace, rail vehicles, and other fields. Inclusions and gases in the melt during the casting process have a great impact on the quality of the ingot. Therefore, in order to prepare high-quality 7085 aluminum alloy ingots, this study used a self-developed ultrasonic-assisted online purification device. The influence of ultrasonic bending vibration on the launder melt inclusions in the semi-continuous casting process of 7085 aluminum alloy was studied, and the following research conclusions were drawn.

(1)In this study, ultrasonic bending vibration is introduced into the launder melt during the semi-continuous casting of 7085 aluminum alloy. Through metallographic analysis and SEM analysis, it was found that there are a large number of long strips with a size of more than 50μm in the melt sample before ultrasonic treatment. However, no large-sized inclusions were found in the melt sample after ultrasonic treatment, and the number of inclusions was small and all of them were smaller than the 30μm point-like small inclusions, which were quantitatively analyzed by ImageJ image analysis software. Ultrasonic bending vibration reduced the average area ratio of inclusions in the melt sample from 3.835 (±0.05)% to 0.458 (±0.05)%, and the purification effect was remarkable.(2)The EDS energy spectrum analysis of aluminum melt specimens revealed that the inclusions of 7085 aluminum alloy melt were mainly composed of oxides, and refining agent component elements such as Cl, Na, Ca, and Rb were found in the inclusions of the melt specimens before ultrasonic treatment, but not in the inclusions of the melt specimens after ultrasonic treatment, indicating that ultrasonic bending vibration can effectively remove the 7085 aluminum alloy melt the residual refining agent can avoid the liquid residual refining agent from entering the crystallizer melt and solidifying to form inclusions defects.(3)Further analysis of the residual aluminum melt specimen and the inner surface of the ceramic cavity channel of the ultrasonic purification device revealed that residual refining agent component elements such as Cl, K, Na, and Ca exist in the edge area and the outer surface of the residual aluminum melt specimen of the ceramic cavity channel, as well as inclusions on the inner surface of the ceramic cavity channel, indicating that under the action of ultrasonic bending vibration, the tiny inclusions in the melt aggregate with each other and combine with the residual refining in the melt. The agent acts to further grow, and it floats or sinks in the ceramic cavity channel and moves to the inner surface of the ceramic cavity to attach to the inner surface of the ceramic cavity, so as to be effectively purified and removed.

## Figures and Tables

**Figure 1 materials-15-03598-f001:**
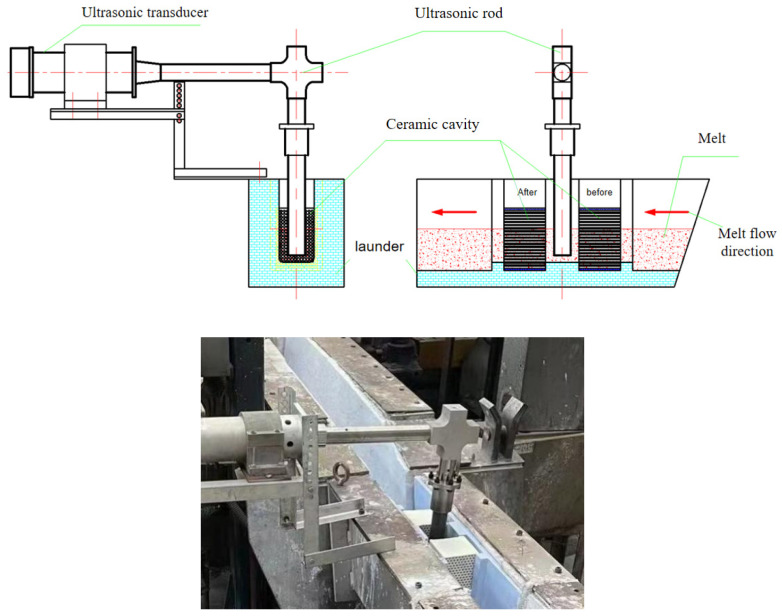
Two-dimensional image and physical photos of the online ultrasonic purification device for launder melt.

**Figure 2 materials-15-03598-f002:**
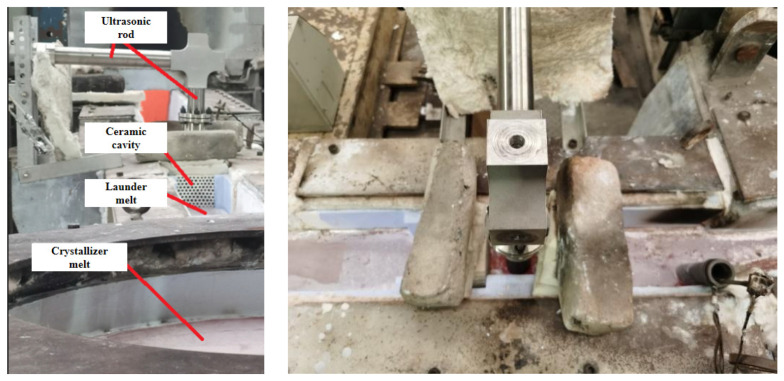
Online ultrasonic purification of launder melt site photo.

**Figure 3 materials-15-03598-f003:**
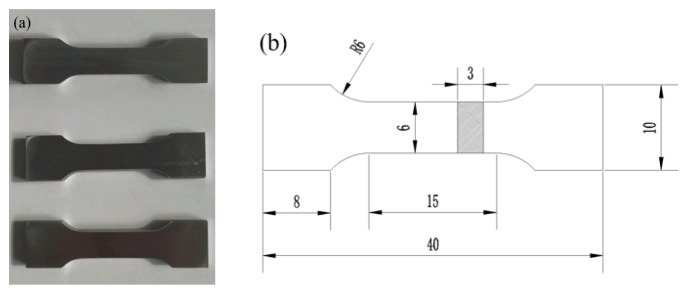
Schematic diagram of the tensile test specimen: (**a**) physical drawing, (**b**) dimension drawing.

**Figure 4 materials-15-03598-f004:**
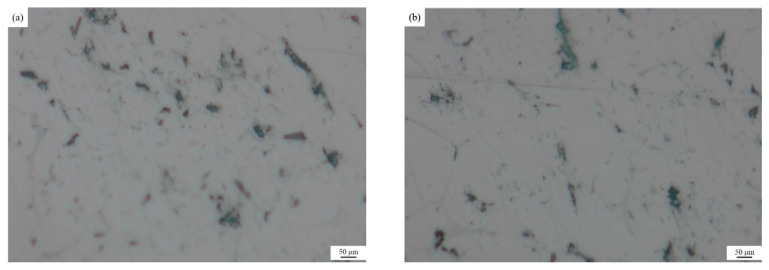
Metallographic pictures of the melt samples before and after the purification device when the ultrasonic wave is not turned on: (**a**) Before the purification device (**b**) After the purification device.

**Figure 5 materials-15-03598-f005:**
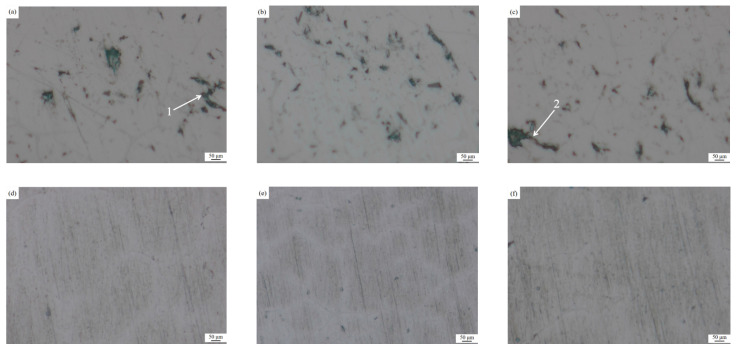
Metallographic pictures of melt specimens before and after ultrasonic purification treatment: (**a**) Specimen 1 (before treatment) (**b**) Specimen 2 (before treatment) (**c**) Specimen 3 (before treatment) (**d**) Specimen 4 (after treatment) (**e**) Specimen 5 (after treatment) (**f**) Specimen 6 (after treatment).

**Figure 6 materials-15-03598-f006:**
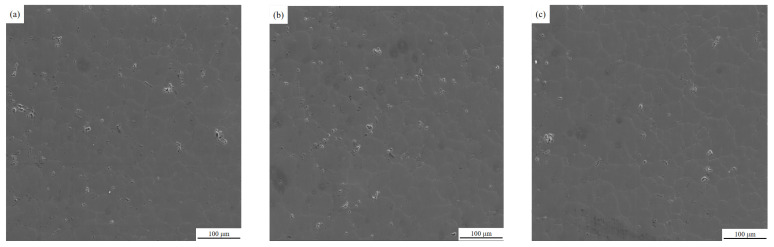
SEM photographs of melt specimens before and after ultrasonic purification treatment: (**a**) Specimen 1 (before treatment) (**b**) Specimen 2 (before treatment) (**c**) Specimen 3 (before treatment) (**d**) Specimen 4 (after treatment) (**e**) Specimen 5 (after treatment) (**f**) Specimen 6 (after treatment).

**Figure 7 materials-15-03598-f007:**
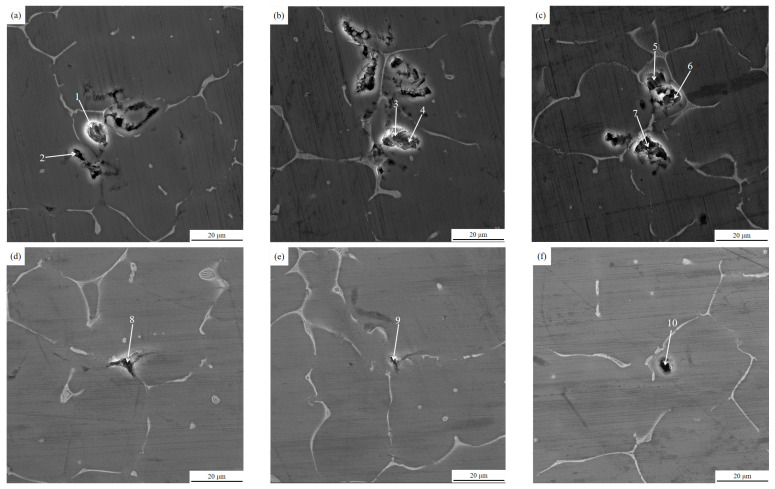
Inclusions in melt specimens before and after ultrasonic purification: (**a**–**c**) for specimens (before treatment), (**d**–**f**) for specimens (after treatment).

**Figure 8 materials-15-03598-f008:**
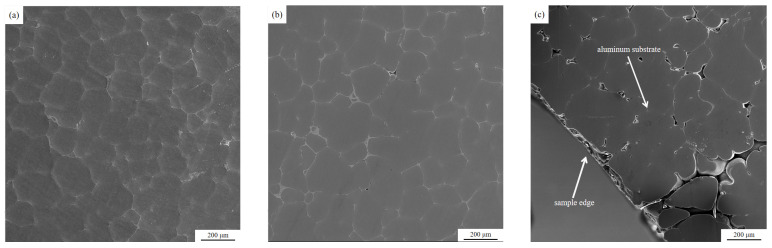
SEM image of the residual aluminum melt sample inside the former ceramic cavity: (**a**) central area, (**b**) transition area, (**c**) marginal area.

**Figure 9 materials-15-03598-f009:**

SEM image of the residual aluminum melt sample inside the post-ceramic cavity: (**a**) central area, (**b**) transition area, (**c**) marginal area.

**Figure 10 materials-15-03598-f010:**
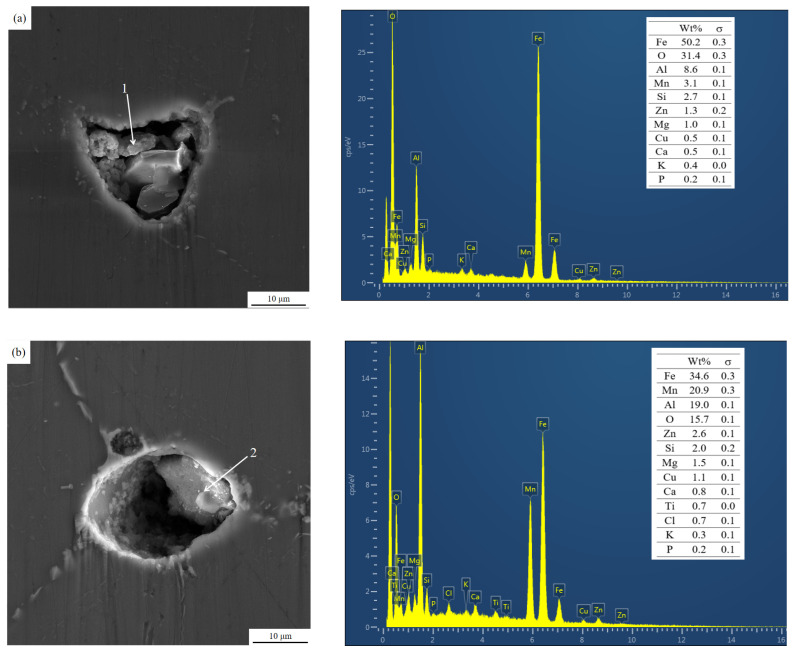
Inclusions contained in residual aluminum melt samples in ceramic cavity: inclusion morphology and EDS spectrum. (**a**) Inclusions 1/(**b**) Inclusions 2.

**Figure 11 materials-15-03598-f011:**
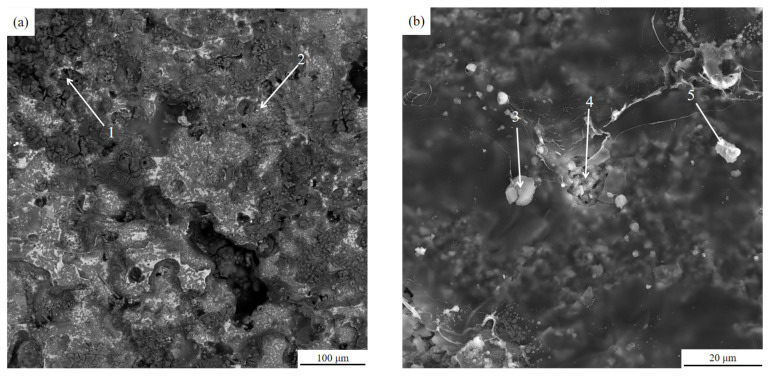
Surface (**a**) and inner surface of cavity channel (**b**) micromorphology of residual aluminum melt sample in ceramic cavity.

**Figure 12 materials-15-03598-f012:**
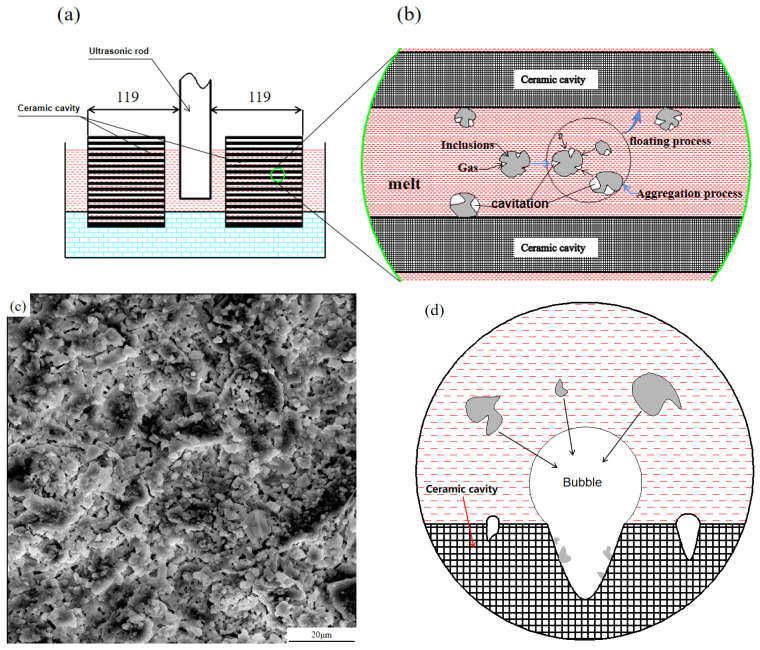
Ultrasonic purification process of inclusions: (**a**,**b**) cavitation purification principle, (**c**) microscopic morphology of the inner surface of the ceramic cavity, (**d**) principle diagram of adsorption of inclusions on the inner surface of the ceramic cavity.

**Table 1 materials-15-03598-t001:** Chemical composition of 7085 aluminum alloy (wt%).

Composition	Zn	Mg	Cu	Zr	Fe	Si	Mn	Al
Content	7.8	1.6	1.8	0.12	0.08	0.03	0.10	Bal.

**Table 2 materials-15-03598-t002:** Content of each chemical component of inclusions in the melt before and after ultrasonic treatment (wt%).

Test Location	O	Mg	Al	Cu	Zn	Cl	Rb	Na	Ca	Si
1	5.18	1.25	82.90	1.2	9.47					
2	1.70	0.70	83.70	1.30	11.30	0.20	1.10			
3	1.92	1.21	79.20	2.45	14.49			0.73		
4	6.40	1.1	60.3	3.5	13.1					15.6
5	9.9	4.4	50.6	13.7	21.1	0.30				
6	11.4	1.0	75.3	1.6	10.3				0.40	
7	3.0	1.3	82.5	1.8	11.4					
8	3.93	1.32	83.54	2.39	8.82					
9	4.43	1.16	76.55	3.56	14.30					
10	2.9	1.3	82.5	2.4	10.9					

**Table 3 materials-15-03598-t003:** Surface composition of the residual aluminum melt sample in the ceramic cavity and the inner surface of the cavity channel (wt%).

Test Location	O	Mg	Al	Cl	Ti	Cu	Zn	K	Na	Si	Zr	Nb	Fe	Pt	Ca
1	56.41	0.96	38.54	0.52			3.12	0.47							
2	52.05	17.57	15.65	0.17			12.38	0.26	1.68						
3	46.96	22.11	10.78	1.59	12.78	0.34	0.83			1.58	1.26	1.15			
4	30.76	1.10	44.80		1.06			0.41		15.86		1.63	1.47	2.91	
5	38.31	13.41	34.19		1.3			0.48		10.37			1.65		0.28

**Table 4 materials-15-03598-t004:** Comparison of tensile mechanical properties of aluminum melt solidified samples in launders.

Specimen Type	Tensile Strength/MPa	Yield Strength/MPa	Elongation/%
Before ultrasonic purification	210.14	209.87	2.6
After ultrasonic purification	230.02	218.62	4.1

## Data Availability

Not applicable.

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
