# Peer review of "Ultrasonic Bending Vibration-Assisted Purification Experimental Study of 7085 Aluminum Alloy Melt"

_materials, 2022, doi:10.3390/ma15103598_

Round 1

Reviewer 1 Report

This paper focuses on the industry-interested process for high-strength Al alloy purification. After reading through it, I must reject this paper for the following reasons: Writing is indeed unsatisfactory, and some terminologies are hard to understand; no solid introduction and background information for why casting is needed for 7xxx Al alloy; The results are less reliable (as common EDS can not capture C accurately) and not complete (no mechanical property comparison AT ALL!!!); and the discussion is scientifically wrong (particularly for the bubble pressure…)

With this, I would reject this paper.

  1. The English is really hard to understand…e.g., what is “Online purification effect”? Is there “offline effect”?
  2. Indeed, 7xxx alloys are less castable, because they are highly susceptible to cracking (e.g., hot cracking issues). Since this paper is discussing about casting AA7085 alloy, ref.s and literatures are needed to prove that casting is feasible for 7xxx, and some of them are shown below and should be included:

[1] Nanotreating High-Zinc Al–Zn–Mg–Cu Alloy by TiC Nanoparticles. J Yuan, M Zuo, M Sokoluk, G Yao, S Pan, X Li. Light Metals 2020, 318-323

[2] Nanoparticles enabled mechanism for hot cracking elimination in aluminum alloys. M Sokoluk, J Yuan, S Pan, X Li. Metallurgical and Materials Transactions A 52 (7), 3083-3096

  1. Table 2: This is EDS results? Why EDS can capture C element? Besides, that much carbon content is clearly not possible…
  2. In the discussion (page 12) “the pressure in the bubbles is lower than that of the melt”…This is scientifically wrong! To sustain the bubble shape, the Laplace pressure must play a role, which means that the pressure is generally higher inside the bubble, given the normal bubble shape…

This means that the whole discussion of this paper is not scientifically valid…

  1. For Al alloy, the performance is not only related with microstructure. Please provide the mechanical property comparison (especially tensile property) for AA7085 samples with and without this new method.

Author Response

Thank you very much for your comments on our manuscript, we have made corresponding improvements.Please see the attachment

Reviewer 2 Report

The authors of the paper “Ultrasonic bending vibration-assisted online purification experimental study of 7085 aluminum alloy melt" have investigated the microstructure of the high-strength aluminum alloy after ultrasonic treatment. The decrease in the non-metallic inclusions amount was shown. However, the reason for such changes is not clear because of not correct experiments (please, see comment #2). Some other points in the paper are also questionable:

  1. The scientific novelty of the paper is unclear. Shi et al already applied the ultrasonic treatment to the aluminum melts (Refs 19, 20, 22). They have already shown the positive influence of the L-shaped ultrasonic rod on the microstructure and mechanical properties. What is new in their current work?
  2. The authors provide the analysis of the microstructure of the samples that were cut from two parts of the casting system: before ceramic cavity and after. In my opinion, it is not correct methodologically. It is hard to recognize the influence of ultrasonic treatment on melt purification. Because the influence of the ceramic cavities themselves on the melt purification in the samples before the ceramic cavity do not consider. The authors should make more pure experiments to investigate the microstructure of the samples after the ceramic cavity without ultrasonic treatment. It is more correct to compare such samples to recognize the true effect of the ultrasonic treatment on melt purification.
  3. The results of the EDS analysis do not have any valuable meaning. The chemical composition of the particles is significantly different from each other and cannot provide any scientific results. The size of the electron beam diameter is larger than some of the analyzed inclusions. As a result, the influence of the aluminum matrix is included in the “chemical composition” of the particles. At the same time, the low values of the concentration of some elements are out of the accuracy of the EDS analysis. I recommend removing Tables 2-4 and correspondent text.
  4. Minor corrections:
  • It is better to add standard deviations to the values of the microstructural characteristics such as inclusion size and its volume fraction.
  • The language of the manuscript should be improved.

Author Response

(The authors gave the same response as above.)

Reviewer 3 Report

The authors did a good job on revealing the ultrasonic bending vabration-assisted online purification study of an Al alloy. Some suggestions are provided to further improve the manuscript.

1) For sample prepartion in 2.3, please provide some more details such as the  etchant, the sand paper grade for granding.

2) For figure 3 and 4, I am confused by the name specimen 1,2,3,4,5,6. What's the difference between the 1,3,5? Do you consider the influence of treatment conditions? I think specimen 1,2 is a pair, so as specimen 3,4 and 5,6, do the specimens in the pair describe the same site?

3) In Table 2, which specimens belong to those before treatment, and which belong to after?

4) For some figures such as Figure 6 and 7, the subtitle is used in the Figure, while not in the caption. Please describe the subtitle also in the caption.

5) For scientific rigour, it's better to provide some EDS spectrums, rather than only tablized data.

Author Response

(The authors gave the same response as above.)

Reviewer 4 Report

  • At Section 2.3., specify the version of software used in this study (line 119)
  • For all instruments (e.g., metallographic microscope, automatic scanning electron microscope), specify the manufacturer, city and country (lines 117, 120).
  • At Section 4, line 265, the parameter “p” is not mentioned. Specify, what represent this parameter.
  • At Section 4, modify “Fig. 10(b)” with “Figure 10(b)” (line 291)
  • At Section 4, modify “Fig. 10(d)” with “Figure 10(d)” (line 300)
  • At conclusions, specify the sectors of the industry were these materials can be used.
  • Correct the References using the Guide of the Journal.

Author Response

(The authors gave the same response as above.)

Round 2

Reviewer 1 Report

The authors have carefully revised the discussion for Introduction, added mechanical tests, and refined/polished their results and discussion.

I'm happy that my concern is addressed point to point. 

Reviewer 2 Report

The authors have answered the previous comments and improved the manuscript. The paper may be accepted for publication.

Reviewer 3 Report

I am satisfied with the response.

Reviewer 4 Report

Dear Sirs,

The manuscript was improved and it can be publish in this form.